# The Potential of Soluble Proteins in High-Moisture Soy Protein–Gluten Extrudates Preparation

**DOI:** 10.3390/polym15244686

**Published:** 2023-12-12

**Authors:** Meng Ning, Yan Ji, Jinchuang Zhang, Hongyang Pan, Jie Chen

**Affiliations:** 1School of Mechanical Engineering, Jiangnan University, Wuxi 214122, China; 8201602011@jiangnan.edu.cn; 2School of Food Science and Technology, Jiangnan University, Wuxi 214122, China; 6200112031@stu.jiangnan.edu.cn (Y.J.); chenjie@jiangnan.edu.cn (J.C.); 3Institute of Food Science and Technology, Chinese Academy of Agricultural Sciences, Beijing 100193, China; zhangjinchuang1002@163.com; 4Analysis Centre, Jiangnan University, Wuxi 214122, China

**Keywords:** soluble proteins, SPI-WG blends, high-moisture extrusion, anisotropic index, rheological properties, permeability

## Abstract

In this study, the effects of different soluble proteins, including collagen peptides (CP), soy protein hydrolysate (HSPI), whey protein isolate (WPI), sodium caseinate (SC), and egg white protein (EWP), on the structural and mechanical properties of blends containing soy protein isolate (SPI) and wheat gluten (WG) were investigated using high-moisture extrusion. The addition of CP and HSPI resulted in a more pronounced fibrous structure with increased voids, attributing to their plasticizing effect that enhanced polymer chain mobility and reduced viscosity. WPI, SC, and EWP acted as crosslinking agents, causing early crosslink formation and decreased polymer chain mobility. These structural variations directly influenced the tensile properties of the extrudates, with CP displaying the highest anisotropic index. Moreover, the presence of soluble proteins impacts the permeability of the extrudates. These insights shed light on how soluble proteins can be used to modify the properties of SPI-WG blends, making them suitable for meat analogue production.

## 1. Introduction

In the current landscape of food production, the dynamic interplay of health, ecological, and ethical concerns has spurred significant shifts in consumer dietary preferences [1]. Notably, plant-based proteins have gained substantial popularity among consumers, reflecting a growing inclination toward more sustainable choices. A key response to these preferences is the development of meat analogues, as consumers highly value the texture and juiciness of meat [2,3]. High-moisture extrusion (HME), owing to its efficiency, cost-effectiveness, low pollutant emission, and the similarity of the extrudate fiber to real meat, has been emerged as one of a highly promising techniques in the development of plant protein-based meat analogues [4,5]. There has been extensive application in the preparation of meat alternatives, leveraging a variety of plant proteins, such as soybean protein, pea protein, peanut protein, and wheat gluten [6,7]. HME allows for the creation of products with a moisture content similar to real meat, eliminating the need for a rehydration step in the production process of meat analogue compared to low-moisture extrusion. The extrudate can be directly marinated to impart flavor or adjust juiciness, resulting in the production of meat analogue.

During the extrusion process, biopolymer chains undergo a series of transformations, including denaturation, dissociation, unravelling, and alignment. It is important to note that an excess of covalent crosslinking within the protein structure can reduce chain fluidity, increase viscosity, and detrimentally impact the formation of fibrous structures in the extruded products [8]. To mitigate this issue, the introduction of plasticizers to the protein system is a well-acknowledged strategy [9,10]. Plasticizers disperse around protein molecules, effectively increasing the spacing between protein–protein interactions. This elevated molecular fluidity results in a reduced friction coefficient, viscosity, and a decrease in the glass temperature (Tg) or flow temperature (Tf). Consequently, the incorporation of plasticizers is conducive to the generation of multiphase states within the system, facilitating the formation of fibrous formation [11,12].

However, it is essential to strike a balance in the use of plasticizers, excessive amounts can cause their independent separate phase, rendering them ineffective in their role as polymer plasticizers [13]. In the realm of plasticizers, various options can result in varying degrees of polymer phase separation during extrusion, resulting in different fibrous structures of extrudates. Among these, water emerges as one of the most potent plasticizers for biopolymer materials. Chen (2010) [14] demonstrated that the extrudates with moisture content as high as 60% exhibited enhanced toughness, chewiness, adhesiveness, and improved fibrous structure as the moisture content increased from 28% to 70%. Palanisamy (2019) [15] has indicated that both excessively low (40%) and exceedingly high (68%) water feed are counterproductive for the formation of fibrous network structures in extruded lupin protein. Apart from water, proteins with low molecular weight or protein hydrolysates have garnered attention as effective plasticizers. Wang (2023) [15] showed the successful extrusion of soy protein–hydrolyzed wheat gluten blends, resulting in the acquisition of fibrous appearances and mechanical anisotropy in extrudates. Similarly, Ji (2023) [16] demonstrated that the addition of hydrolyzed soy protein isolate (HSPI) at low concentrations enhanced fibrous appearances and mechanical anisotropy, while higher concentrations lead to compact and brittle structures. Hoyos-Conch (2022) [17] revealed that the addition of high protein hydrolyzed during extrusion can reduce the specific mechanical energy (SME), yielding extrudates characterized by elevated durability and reduced hardness.

In addition to promoting the formation of fibrous structures, the inclusion of plasticizers has been observed to enhance the incorporation of air bubbles within the protein matrix [12]. Although the presence of air is not a prerequisite for fiber formation, it appears to improve both fibrous appearance and mechanical anisotropy of the extrudate. Moreover, the porous structure proves advantageous in facilitating the penetration of marinades during the production of meat analogues, positively contributing to their juiciness [11,18]. Our previous study revealed that the inclusion of 10% soluble WG hydrolysate within a blend with 70 wt% SPI and 30 wt% WG resulted in notable enhancements in the anisotropic properties of extrudates [19]. It is expected that different soluble proteins will enable the achievement of the most suitable plasticizing effect, leading to enhanced attributes in extrudates regarding their fiber formation, tensile strength, and permeability. We chose five model proteins (collagen peptide (CP), hydrolyzed soy protein isolate (HSPI), whey protein isolate (WPI), sodium caseinate (SC), egg white protein (EWP)). These proteins were chosen based on their high solubility and relatively small molecular weight, akin to WG hydrolysate used in our prior study [19]. Their solubility, sodium dodecyl sulfate–polyacrylamide gel electrophoresis, and molecular weight distribution were measured. Additionally, we determined the viscoelastic properties of SPI-WG blends with varied soluble proteins through time sweep and temperature sweep tests. The extrudates were produced in an extruder using blends consisting of 70 wt% SPI, 20 wt% WG, and 10 wt% soluble proteins, while a control sample was made with 70 wt% SPI and 30 wt% WG.

## 2. Materials and Methods

### 2.1. Materials

Soy protein isolate (SPI, SD-100) was purchased from Linyi Shansong Biological Products Co., Ltd. (Linyi, China). The SPI powder contains 90 wt% protein, 7.0 wt% water, 0.8 wt% fat, and 2 wt% carbohydrate. Wheat gluten (WG) was purchased from Zhongyu Food Co., Ltd. (Binzhou, China) with 7.7 wt% water. WG contains 85.6 wt% protein, 0.9 wt% ash, and 0.8 wt% fat on a dry matter basis according to the manufacturer’s specifications. Commercial hydrolyzed soy protein isolate (HSPI, 791) was obtained from Suizhong Song Zhiyuan Plant Protein Co., Ltd. (Huludao, China). Sodium caseinate (SC) was purchased from the Fonterra Co-operative Group Ltd. (Auckland, New Zealand). Collagen peptide (CP) was purchased from Jiangsu Taidu Bioengineering Co., Ltd. (Nanjing, China). Egg white protein (EWP, RD1010) and whey protein isolate (WPI, 9410) were purchased from Shanghai Yongrui Biological Co., Ltd. (Shanghai, China) and Jiangsu Qianbo Bioengineering Co., Ltd. (Nanjing, China), respectively. All other chemicals utilized in this study were of analytical grade unless specified otherwise.

### 2.2. Solubility

The dispersion (2 wt%) of protein was centrifuged at 10,000× *g* for 30 min at 20 °C, and the supernatant was discarded [20]. The mass of the dry pellet was also recorded after freeze-drying. The solubility was determined using Equation (1):(1)$$Solubility\;\left(\%\right)=\frac{M_{dry\;powder}-M_{dry\;pellet}}{M_{dry\;powder}}$$
where $M_{dry\;powder}$ represents the mass of the overall added dry powder and $M_{dry\;pellet}$ is the mass of the pellet after centrifugation and drying.

### 2.3. Sodium Dodecyl Sulfate–Polyacrylamide Gel Electrophoresis

Sodium dodecyl sulfate–polyacrylamide gel electrophoresis (SDS-PAGE) was carried out on a discontinuous buffered system with a 12% separating gel and a 4% stacking gel. Samples (4 mg/mL) were diluted (1:1, *v*/*v*) in buffer containing 0.0625 M Tris-HCl, 10% glycerine, 2% SDS and 0.0025% bromophenol blue with and without β-mercaptoethanol (5%, *v*/*v*). The prepared samples (15 μL) were loaded onto the gels and the electrophoresis was run at 80 V and then at 120 V. The eluted gels were photographed using a gel imaging system (version 5.2; 2104 Bio-Rad Laboratories, Irvine, CA, USA).

### 2.4. High-Performance Liquid Chromatography

High-performance liquid chromatography (HPLC) was conducted using an HPLC system (Waters 2695, Milford, MA, USA) equipped with a Shodex protein KW-804 column (8 mm inner diameter × 30 cm, Shodex Co., Tokyo, Japan). The samples were diluted to 10 mg/mL with deionized water and filtered through a 0.45 μm film. The elution buffer was 0.05 mol/L phosphate buffer (pH 7.0) containing 0.03 mol/L sodium chloride. The elution rate was set at a flow rate of 1 mL/min, and a 10 μL injection volume was used for analysis. The detection wavelength was set at 220 nm. Molecular weight (MW) standards included aprotinin (MW 6.5 kDa), phosphatase (MW 32 kDa), ovalbumin (MW 43 kDa), bovine serum albumin (MW 67 kDa), and bovine thyroglobulin (MW 669 kDa).

### 2.5. Rheological Properties of Protein Blends

A closed cavity rheometer (RPA Elite, TA Instruments, New Castle, DE, USA) was used to assess the reactions of blends under different thermal and mechanical stresses [21,22]. The test sample consisted of SPI, WG, and soluble protein in a weight ratio of 7:2:1. A 45 wt% SP-WG-soluble protein mixture was prepared and hydrated at 8 °C for at least 24 h. Approximately 5 g was placed in a closed chamber between two plastic films. Sealing the cones prevents water evaporation at high temperatures so that a cavity pressure of 4.5 MPa can be achieved. Mechanical treatment was provided by the oscillatory movement of the lower cone and can be adjusted by varying the angular frequency ω and strain γ.

First, an oscillation time sweep experiment was conducted at a high frequency (10 Hz) and strain (80%) for 15 min at various temperatures (100–140 °C). Second, a temperature sweep experiment (10 Hz, 80%) was performed from 40 °C to 140 °C at a heating rate of 5 K/min.

### 2.6. Preparation of Samples during the Extrusion Process

A twin-screw extruder with co-rotating design (FMHE36-32, FUMACH, Changsha, China) was used to prepare the extrudates. The screws were with a 36 mm screw diameter and a 32 length-to-diameter ratio. The extruder borrel was divided into 8 separately heated and thermally controlled zones. A water pump was employed to introduce water into the extruder. Appendix A illustrates the screw configuration elements from the feeding point to the exit die. Throughout the trials, the screw configuration consisted of both forward and reverse transport elements. The protein mixture was prepared by mixing SPI, WG, and one of other proteins at a ratio of 7:2:1. The control sample was prepared with SPI and WG at a ratio of 7:3. The extruder barrel consisted of eight sections, the first one included the gate for solid feed without the temperature control, and the eighth section had an individual temperature control. Based on preliminary experiments, the extrusion conditions were set as follows: a feed moisture content of 55 wt% (dry basis), the screw speed of 280 rpm, and the extruder barrel temperatures of 60 °C, 80 °C, 110 °C, 130 °C, 140 °C, 140 °C, and 110 °C from the second zone to the eighth, respectively. The end of the extruder was fitted with a cooling die with dimensions of 10 × 70 × 926 mm (H × W × L) and was maintained at 70 °C by running water. More detail about the extruder can be seen in Deng’s paper [23]. During steady-state operating conditions, extrudates were collected and designated as E-Control, E-CP, E-HSPI, E-WPI, E-SC, and E-EWP. All extrudates were stored in polyethylene bags in a freezer (−18 °C) before further analysis.

### 2.7. System Response Parameters

The die pressure and torque during the extrusion processing were automatically recorded every 10 s by an on-line system. The specific mechanical energy (SME) was calculated according to the equation as follows:(2)$$SME\left(\mathrm{kJ}\cdot {\mathrm{kg}}^{-1}\right)=\frac{2\pi \times n\times T}{MFR}$$
where n (rpm) was the screw speed, T (N∙m) was the motor torque, and MFR (g∙min^−1^) was the mass flow rate.

### 2.8. Tensile Strength Analysis

A texture analyzer (TA.XT2, Stable Micro Systems, Ltd., Godalming, UK) was used with a load cell of 100 N. A dog bone-shaped mold was used to cut extrudates in parallel or perpendicular to the extrusion direction [23]. The samples were approximately 15.2 mm in length, 3.18 mm in width, and varied in thickness between 4 and 6 mm. The test was conducted at room temperature with a displacement rate of 3 mm/s [6]. Abrasive paper was used to prevent slipping during testing when the samples were placed between two sand-coated clamps at a distance of 15.2 mm. A stress–strain curve was used to determine tensile stress and tensile strain at rupture. For each sample, a total of 9 specimens in parallel and perpendicular directions were collected, and the data were averaged with standard deviations computed in accordance with the variation among the six kinds of samples.

### 2.9. Confocal Scanning Laser Microscopy

A confocal laser scanning microscope was used to visualize the microstructures of the extrudates. The samples were cut in parallel and perpendicular to the direction of extrusion (approximately 20 mm) and then rapidly pre-frozen with a freeze-embedding agent (Tissue-Tek O.C.T Compound, Sakura Finetek, Torrance, CA, USA) at −80 ℃. Subsequently, the samples were placed on a tray and sectioned with a cryo-microtome (CM1950, Leica, Berlin, Germany) at −20 °C, yielding approximate 20 μm thick slices in a regular pattern. These slides were then affixed to a pretreated slide and stored at −20 °C in the refrigerator. To stain the proteins, Rhodamine B was used at a concentration of 0.2 mg/mL. Visualization was conducted using a confocal scanning laser microscope type TCS SP8 (Leica, Germany) with an OSPL laser at 552 nm, and images were captured with an HC PL APO 10×/0.40 CS lens. At least 10 images were analyzed for each sample.

### 2.10. Bulk Density

The bulk density of the extrudates (g/mL) was determined by dividing their weight by volume. The volume of the extrudates was measured by displacement of rape seed [24].

### 2.11. Permeability Measurement

Permeability of the extrudates was assessed using the following method: The extrudates were precisely cut into sizes of 2 cm × 2 cm and then carefully placed into vacuum pouches. Subsequently, the extrudates were marinated for 10 min with a 5‰ solution of cochineal red pigment, employing a household vacuum packing machine (C010, Multivac, Berlin, Germany) at a vacuum pressure of 15 bar. Once marination was complete, the samples were cut in both parallel and perpendicular directions. The permeability was defined as the mass of the extrudates absorbing the marinade divided by the original mass of the extrudates. The experiment was repeated six times for each group of samples.

### 2.12. Statistical Analysis

All measurements were carried out in triplicate unless otherwise specified. Analyses were conducted using Statistic version 9.0 (Analytical Software, Tallahassee, FL, USA). ANOVA with the least significant difference and 95% confidence interval was used to compare the means.

## 3. Results and Discussion

### 3.1. Physicochemical Properties of Protein

#### 3.1.1. Solubility

The solubility of the selected proteins is depicted in Figure 1. Collagen peptide (CP) and whey protein isolate (WPI) exhibited near-complete solubility. Sodium caseinate (SC) displayed a solubility of 91.0%, while egg white protein (EWP) and hydrolyzed soy protein isolate (HSPI) revealed solubilities of 80.7% and 67.7%, respectively.

#### 3.1.2. Molecular Weight Distribution

The SDS-PAGE patterns of five soluble proteins under both reducing and non-reducing conditions are shown in Figure 2A,B. CP had a molecular weight smaller than 14.4 kDa. HSPI exhibited bands mainly under 14.4 kDa, with a few bands ranging from 14.4 to 31 kDa. WPI primarily contained α-lactalbumin (14.2 kDa) and β-lactoglobulin (18.4 kDa). SC displayed bands around 31 kDa, composing α-, β-, and κ-caseins. EWP mainly contained ovalbumin (43 kDa), ovo transferrin (76 kDa), and lysozyme (14.3 kDa). When subjected to reducing conditions, significant bands remained evident at the top of gel for SC and EWP, indicating a dominant role of non-covalent bonds in protein aggregation.

The hydrodynamic molecular weight of soluble proteins was characterized with size exclusion HPLC (Figure 2C). The profiles of soluble proteins displayed a single peak, except for WPI. The retention time of the peak followed this order: CP > HSPI > WPI > SC > EWP. This indicated that the molecular weight of those proteins increased in the opposite order, aligning with the SDS-PAGE results.

#### 3.1.3. Viscoelastic Properties of Blends

The changes in dynamic rheology of SPI-WG blends with varied soluble proteins during heating at 40–140 °C and cooling at 140–70 °C were monitored (Figure 3). The complex modulus (G′) of blends with CP, HSPI, and SC closely resembled the control blend at 40–85 °C, rising to varied extents at 85–110 °C before descending again at higher temperatures (Figure 3A). The value of tan δ decreased with increasing temperature at 85–110 °C and increased again at higher temperatures (Figure 3C). The increased G′ and reduced tan δ at 85–110 °C can be linked to increased molecular weight due to gluten polymerization reactions [6,25], with the control blend showing the highest enhancement in G′ at 85–110 °C. In the case of SC, its tan δ continued to decrease until 135 °C, which is probably due to the aggregation of SC at elevated temperature [26]. Blends with WPI and EWP displayed lower G′ than the control blend at 40–65 °C, but rapidly increased from 65 °C to 75 °C, then remained stable until approximately 110 °C before declining at higher temperature. Their tan δ decreased with increasing temperature and exhibited three stages at 40–65 °C, 65–75 °C, and 75–110 °C, respectively. At 65–75 °C, tan δ sharply decreased from >1 to <1, indicating gel formation due to heat-induced WPI and EWP denaturation [27,28].

During the cooling process (140–70 °C), G′ values of all blends increased with decreasing temperatures (Figure 3B). The tan δ remained constant at 140–95 °C and increased with increasing temperatures at 95–70 °C (Figure 3D). Blends with CP and HSPI exhibited higher tan δ values than those with SC, WPI, and EWP, indicating that CP and HSPI reduced the solid-like behavior during cooling.

Figure 4 presents the apparent elasticity modulus (G′) of SPI-WG blends at each temperature (100 °C, 110 °C, 120 °C, 130 °C, and 140 °C). Blends with WPI and EWP exhibited higher G′ than the control blend over time at all the measured temperatures, due to crosslinking from heat-induced denaturation (Figure 3). Blends with SC consistently exhibited the lowest G′, likely due to the water released from SC during heating [29]. At 100 °C, the G′ of all blends notably increased within the initial 3 min and followed constant, except for SC. At higher temperatures (120–140 °C), heating SC increased aggregation, resulting in increased viscosity [30]. Although SC exhibited distinct behavior compared to blends with WPI and EWP (Figure 3 and Figure 4), their potential impact during extrusion could be similar.

At 110 °C and 120 °C, the G′ of all blends experienced a significant initial increase and a subsequent slight decline over time. The sharp increase in G′ was explained by the aggregation or polymerization of proteins in earlier studies [31], while the decrease in G′ could be due to the protein degradation at a higher temperature [19]. At 130 °C and 140 °C, G′ increased slightly within initial 1 min and then decreased, possibly due to protein degradation at higher temperatures. In addition, blends with CP and HSPI had lower G′ than the control blend at 110–140 °C (Figure 3A and Figure 4), suggesting a plasticizing effect, promoting the slippage of the blends.

### 3.2. System Response Parameters

Table 1 summarizes the system response parameters, including die pressure, torque, and specific mechanical energy (SME). These parameters directly describe the state of the extrusion process and indirectly reflect the flow behavior of protein melt during extrusion [32]. The addition of CP and HSPI in SPI-WG mixtures significantly reduced the die pressure, torque, and SME during extrusion, indicating their plasticizing effect and improved flow behavior. Conversely, the addition of WPI, SC, and EWP increased these parameters, which could be due to protein aggregation during heating, which is consistent with the rheological results (Figure 3 and Figure 4).

### 3.3. Structural Properties of Extrudates

Structured materials were inspected visually by tearing and folding the edge portion of the extrudates. Figure 5 shows the macrostructure of folded extrudates without soluble protein (E-Control) and with different soluble proteins of collagen peptide (E-CP), hydrolyzed soy protein isolate (E-HSPI), whey protein isolate (E-WPI), sodium caseinate (E-SC), and egg white protein (E-EWP). Replacing partial wheat gluten (10% of total blend, dry base) with CP and HSPI yielded a highly fibrous structure with thin isolated fibers. E-Control, E-WPI, E-SC, and E-EWP materials exhibited a layered structure with few separated fibers.

Extrusion involves protein unfolding above their denaturation temperature, which is due to increased polymer mobility from weakened intramolecular bonds. As temperature rises, polymer mobility increases. Going beyond the flow temperature (Tf) yields a low-viscosity material, making it easier to process, though intermolecular bonds and aggregation limit chain mobility. Interactions breaking and forming among polymers create a temporary network. Subsequent cooling solidifies polymer crosslinking within the protein matrix, with a preferred directional orientation. This crosslinking is influenced by plasticizers that soften the material and reduce the crosslinking density by coming between polymer chains [33]. According to the rheological properties and reduced SME input (Figure 3, Figure 4 and Figure 5), CP and HSPI act as plasticizers in SPI-WG blends, increasing molecular mobility and decreasing viscosity. Conversely, WPI, SC, and EWP act as crosslinking agents, reducing polymer chain mobility through early crosslink formation (denaturation) in the mixing zone [34].

Confocal laser scanning microscopy (CLSM) images in Figure 6 revealed that all the extrudates contain spherical voids with varying content. E-CP and E-HSPI contained more voids compared to E-Control and other extrudates. The bulk density of the extrudate is 1.12 g/mL without soluble protein, decreasing to 0.86 g/mL, 0.94 g/mL and 1.00 g/mL when CP, HSPI, and WPI was added (Figure 7). SC and OVA had little influence on the bulk density compared to E-control. The decreased bulk density is due to increased air incorporation in the extrudates, resulting in a porous structure (Figure 6). This is consistent with our previous studies [16,19]. The presence of soluble protein led to a less solid-like behavior in the protein phase (Figure 3D), which is extensible enough to respond to gas pressure but still strong enough to resist collapsing.

### 3.4. Tensile Strength of Extrudates

The addition of soluble proteins affected the structure and porosity of SPI-WG extrudates, which was expected to impact the tensile properties (Figure 8). Tensile tests were performed in parallel and perpendicular to the extrusion flow direction, and their ratio (AI) indicates anisotropy. The fracture stress (σ) and fracture strain (ε) increased with the molecular weight of added soluble proteins. CP had the highest AI values (1.7 for σ and 1.2 for ε). AI values decreased with increasing molecular weight of soluble proteins. It showed that suitable molecular weight of soluble proteins would facilitate fiber structure. Compared to E-Control, E-SC, and E-OVA had lower AI for σ, and E-WPI, E-SC, and E-EWP had lower AI for ε. The increased AI for E-CP and E-HSPI was due to an increase in fracture stress in the parallel direction, rather than a decrease in fracture stress in the perpendicular direction. Thus, we deduced that the mechanical response is influenced by both the anisotropic structure of protein matrix and the presence of air bubbles.

### 3.5. Permeability of Extrudates

The internal cavities of the extrudates affect the permeability and the initial marinade release, potentially enhancing juiciness [16,19,35,36]. Here, we examined the permeability of SPI-WG extrudates using a tracing dye solution to map out the distribution of marinade (Figure 9A). Penetration of dye solution increase in E-CP, E-HSPI, and E-WPI compared to the control sample, while there was no significant difference in E-SC and E-EWP. This corresponded to the quantity of voids incorporated in the extrudates (Figure 5 and Figure 6).

## 4. Conclusions

The addition of soluble proteins, such as collagen peptide (CP) and hydrolyzed soy protein isolate (HSPI), to SPI-WG blends resulted in structural changes characterized by fibrous structures with increased voids. These changes were attributed to the plasticizing effect of CP and HSPI, leading to enhanced polymer chain mobility and reduced viscosity. In contrast, whey protein isolate (WPI), sodium caseinate (SC), and egg white protein (EWP) acted as crosslinking agents, causing early crosslink formation and decreased polymer chain mobility. The variations in structure and porosity directly affected the tensile properties of the extrudates, with CP demonstrating the highest anisotropic characteristics. Additionally, the permeability of the extrudates was influenced by the presence of soluble proteins, impacting the marinade release. These findings provide insights into the role of soluble proteins in modifying the fibrous structure and marinade permeability of SPI-WG extrudates, which have implications for their use in meat analogue production. Moreover, this study can lay a foundation for better seasoning of extrudates, which can promote the better introduction of a different flavor of meat analogue product into the market.

## Figures and Tables

**Figure 1 polymers-15-04686-f001:**
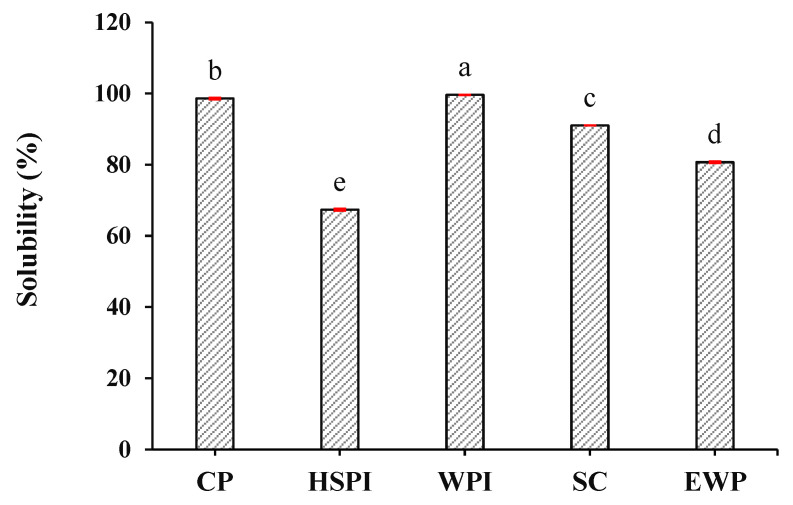
Solubility of selected proteins (collagen peptide (CP), hydrolyzed soy protein isolate (HSPI), whey protein isolate (WPI), sodium caseinate (SC), and egg white protein (EWP). Different letters (a, b, c, d, e) express the significant differences (*p* < 0.05).

**Figure 2 polymers-15-04686-f002:**
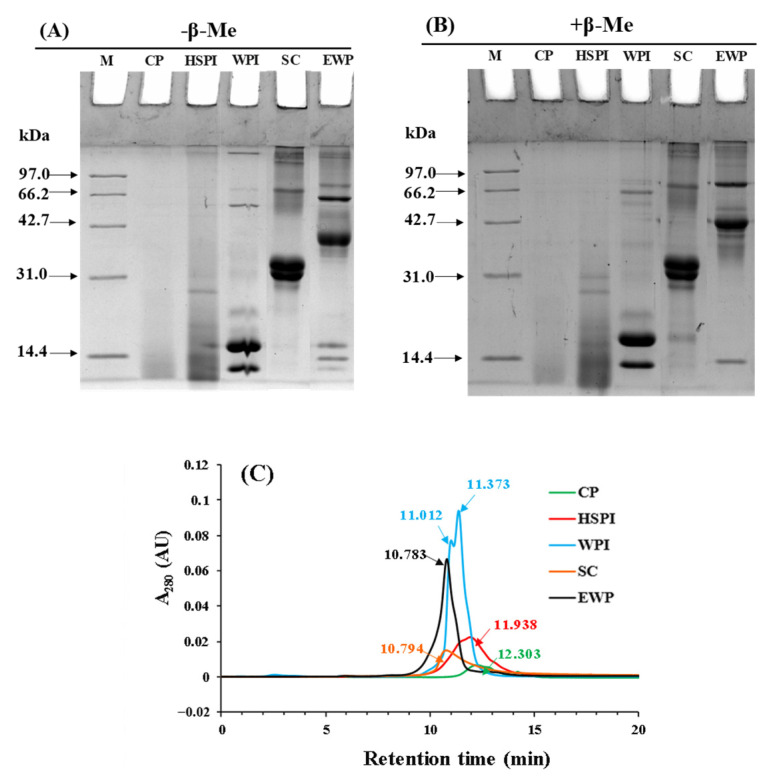
Non−-reduced (**A**) and reduced pattern (**B**) SDS−-PAGE profiles of selected proteins (collagen peptide (CP), hydrolyzed soy protein isolate (HSPI), whey protein isolate (WPI), sodium caseinate (SC), and egg white protein (EWP)); (**C**): size exclusion HPLC profiles of selected proteins. The number indicated the retention time of the peak.

**Figure 3 polymers-15-04686-f003:**
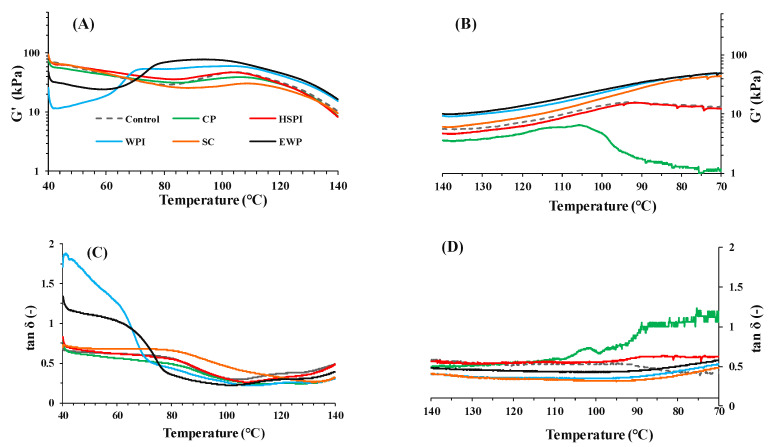
Storage modulus (G′), and loss angle (Tanδ) of SPI-WG blends with varied soluble proteins during heating at 40–140 °C (**A**,**C**) and cooling at 140–70 °C (**B**,**D**).

**Figure 4 polymers-15-04686-f004:**
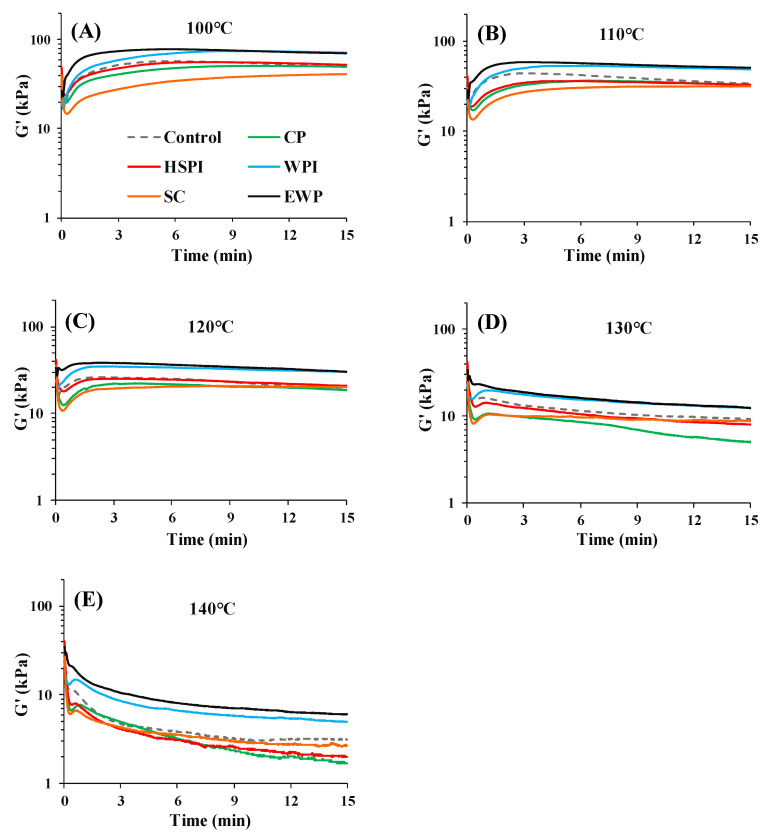
Time sweep measurements (80% strain, 10 Hz frequency) for SPI-WG blends with collagen peptide (CP), hydrolyzed soy protein isolate (HSPI), whey protein isolate (WPI), sodium caseinate (SC), and egg white protein (EWP) at various temperatures: (**A**) 100 °C, (**B**) 110 °C, (**C**) 120 °C, (**D**) 130 °C, and (**E**) 140 °C.

**Figure 5 polymers-15-04686-f005:**
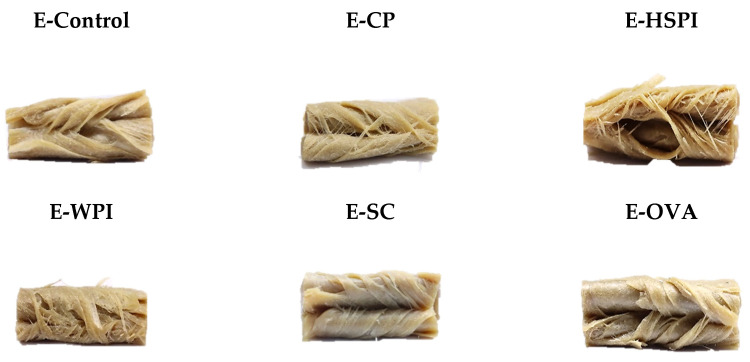
Macrostructure of extruded 45 wt% SPI-WG blends without (E-Control) and with soluble proteins (collagen peptide (E-CP), hydrolyzed soy protein isolate (E-HSPI), whey protein isolate (E-WPI), sodium caseinate (E-SC), and egg white protein (EWP)).

**Figure 6 polymers-15-04686-f006:**
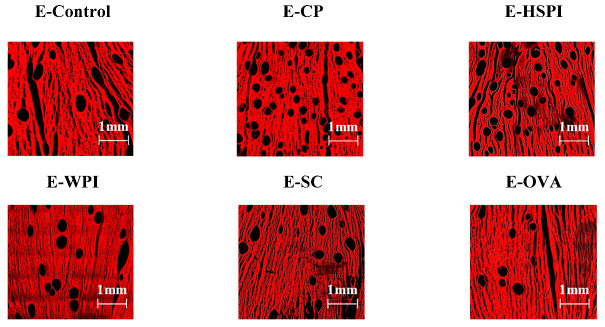
CLSM images of SPI-WG extrudates without (E-Control) and with soluble proteins (collagen peptide (E-CP), hydrolyzed soy protein isolate (E-HSPI), whey protein isolate (E-WPI), sodium caseinate (E-SC), and egg white protein (EWP)).

**Figure 7 polymers-15-04686-f007:**
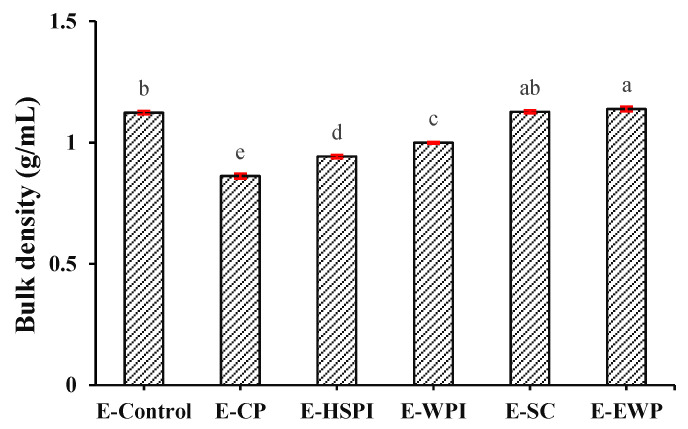
The bulk density of SPI-WG extrudate with varied soluble proteins (collagen peptide (E-CP), hydrolyzed soy protein isolate (E-HSPI), whey protein isolate (E-WPI), sodium caseinate (E-SC), and egg white protein (EWP)). Different letters (a, b, c, d, e) express the significant differences (*p* < 0.05).

**Figure 8 polymers-15-04686-f008:**
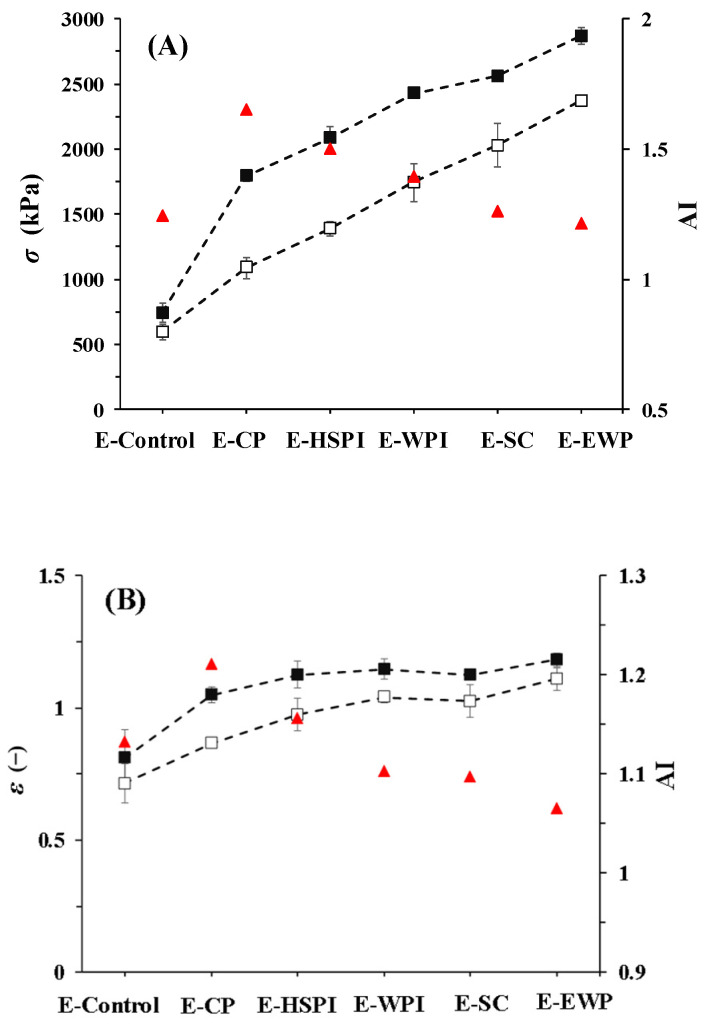
Tensile properties of SPI-WG extrudates deformed in parallel (black square) and perpendicular (white square) directions: (**A**) fracture stress, (**B**) fracture stress with without (E-Control) and with soluble proteins (collagen peptide (E-CP), hydrolyzed soy protein isolate (E-HSPI), whey protein isolate (E-WPI), sodium caseinate (E-SC), and egg white protein (EWP)). The ratio between the average of the parallel and perpendicular directions is a measure of anisotropy (AI, red triangles). The dashed lines guide the eye.

**Figure 9 polymers-15-04686-f009:**
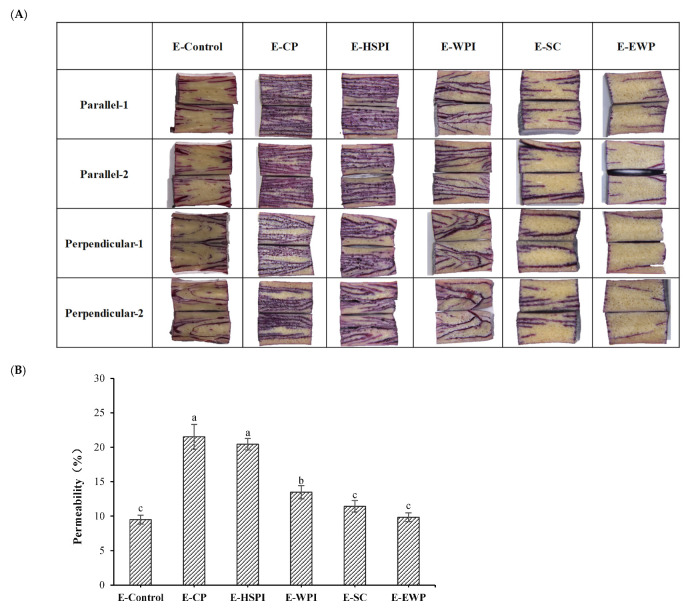
The distribution map (**A**) and permeability (**B**) of 0.5% cochineal red pigment into SPI-WG extrudates. Different letters (a, b, c) express the significant differences (*p* < 0.05).

**Table 1 polymers-15-04686-t001:** Effect of different soluble proteins in SPI-WG mixtures on the system response parameters. Different letters (a, b, c, d, e, f) express the significant differences (*p* < 0.05).

Samples	Die Pressure (MPa)	Torque (N∙m)	SME (kJ∙kg^−1^)
E-Control	1.73 ± 0.01 ^d^	117.21 ± 0.14 ^d^	747.66 ± 0.89 ^d^
E-CP	0.77 ± 0.02 ^f^	113.13 ± 0.07 ^e^	720.01 ± 0.45 ^e^
E-HSPI	1.33 ± 0.10 ^e^	113.45 ± 0.50 ^e^	720.33 ± 3.18 ^e^
E-WPI	2.21 ± 0.00 ^c^	118.67 ± 0.36 ^c^	753.08 ± 2.28 ^c^
E-SC	2.71 ± 0.00 ^b^	120.56 ± 0.23 ^b^	763.67 ± 1.21 ^b^
E-EWP	2.74 ± 0.01 ^a^	124.23 ± 0.19 ^a^	785.99 ± 1.21 ^a^

## Data Availability

The data presented in this study are available on request from the corresponding author.

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
