# Peer review of "The Potential of Soluble Proteins in High-Moisture Soy Protein–Gluten Extrudates Preparation"

_polymers, 2023, doi:10.3390/polym15244686_

Round 1

Reviewer 1 Report

Comments and Suggestions for Authors

Dear Authors,

Overall, this manuscript is interesting and covers an important topic. In my opinion, the manuscript is also written quite correctly. To increase its quality, I have included some important corrections below.

Introduction: At the beginning of the introduction there is no reference to the possibilities of the extrusion process. Mainly in terms of functional food and new applications. First of all, the possibilities of processing food waste raw materials from agri-food processing are important. Check out these works: "Characterization of corn extrudates with the addition of brewers' spent grain as a raw material for the production of functional batters" or "Effect of extrusion on physicochemical properties, functional properties and antioxidant activities of shrimp shell wastes protein". Protein processing is just one option for extrusion (especially proteins with high moisture content).

Introduction: At the end of the introduction, you should include the scientific goal of your research. That's what's missing here. Additionally, you should justify the research you undertake a little better before your goal.

Ch. 2.2. Was the solubility testing methodology your own method? It may be worth referring to the methodology already described in another work.

Ch. 2.6. Describe how to read the temperature profile. E.g. The temperature profile of the cylinder is: 60°C, 80°C, 110°C……….. where 60°C is the dosing section and 110 is the die/head temperature….: Of course, unless I am wrong.

Ch. 2.8. When testing strength, you should refer to the standard.

Ch. 2.11. Is the word "marinated" really correct? Check it.

Figure 1. Are there error whiskers on the graph? If not, plot them on the chart.

Ch. 3.4: Are the results close or far from expected? Write this next to the selected research results. Particularly during strength tests.

Conclusions: The conclusions need to be expanded a bit. Describe what your research will contribute to the development of similar materials. What are the future prospects for this research or these materials?

Author Response

Overall, this manuscript is interesting and covers an important topic. In my opinion, the manuscript is also written quite correctly. To increase its quality, I have included some important corrections below.

Q1Introduction: At the beginning of the introduction there is no reference to the possibilities of the extrusion process. Mainly in terms of functional food and new applications. First of all, the possibilities of processing food waste raw materials from agri-food processing are important. Check out these works: "Characterization of corn extrudates with the addition of brewers' spent grain as a raw material for the production of functional batters" or "Effect of extrusion on physicochemical properties, functional properties and antioxidant activities of shrimp shell wastes protein". Protein processing is just one option for extrusion (especially proteins with high moisture content).

R1Thanks for your valuable suggestion, we cited more references in revised manuscript (Line 32-43). And our research was not focused on processing food waste raw material.

Q2Introduction: At the end of the introduction, you should include the scientific goal of your research. That's what's missing here. Additionally, you should justify the research you undertake a little better before your goal.

R2We have referred the scientific goal of our research in line 85-87. Moreover, our previous study revealed that 10% soluble WG hydrolysate enhanced the anisotropic index of extrudates (Line 83-85). We want to investigate whether other soluble proteins could also be used as plasticizers to enhanced fiber formation, so that the fibers of meat analogues would be closer to real meat fibers.

Q3Ch. 2.2. Was the solubility testing methodology your own method? It may be worth referring to the methodology already described in another work.

R3Thanks for your valuable suggestion, we cited the relevant literature (Line 116).

Q4Ch. 2.6. Describe how to read the temperature profile. E.g. The temperature profile of the cylinder is: 60°C, 80°C, 110°C……where 60°C is the dosing section and 110 is the die/head temperature….: Of course, unless I am wrong.

R4Thanks for your valuable suggestion, we have refined the description of the section (Line 161-163)

Q5Ch. 2.8. When testing strength, you should refer to the standard.

R5Thanks for your valuable suggestion, we cited the relevant literature (Line 186).

Q6Ch. 2.11. Is the word "marinated" really correct? Check it.

R6Yes, "marinated" means soaking something in a sauce/liquid for some time, which was exactly what we used in this study.

Q7Figure 1. Are there error whiskers on the graph? If not, plot them on the chart.

R7We have plotted the error bar on the chart. Because of the better parallelism of the data in this experiment, which causes the error line to look short, we have changed the red color instead (Figure1, 7).

Q8Ch. 3.4: Are the results close or far from expected? Write this next to the selected research results. Particularly during strength tests.

R8: The results close from our expected, we wanted to determine the range of molecular weight of soluble proteins capable of increasing the fiber structure of extrudate. The experimental results showed that suitable molecular weight of soluble protein would facilitate fiber formation of extrudates (Line 352-353).

Q9Conclusions: The conclusions need to be expanded a bit. Describe what your research will contribute to the development of similar materials. What are the future prospects for this research or these materials?

R9Thanks for your valuable suggestion. We rewritten the conclusion in the revised manuscript (Line 389-391).

Reviewer 2 Report

Comments and Suggestions for Authors

The synthesis of polymers derived from a blend of wheat and soy proteins has garthered attention in recent research. Improving these polymers by incorporating small quantities of soluble proteins is a path that has already been explored. A global investigation, as outlined in this article, is both innovative and highly relevant to the scientific community, especially in relation to the enhancement of mechanical properties and anisotropy.

The introduction is clear and the objectives well described. The last pharagraph describes clearly the extruded blends and the control simple. It could be simplified to the obtention different blends made of SPI and WG with 10% of chosen proteins which share similarities with the WG hydrolysate used in a previous study. This 10% correspond to the WG fraction.

Some suggestions from the reviewer are described below:

- Line 71 - The introduction should encompass a description of the requirements or properties that need to be investigated to ascertain the production of a meat analogue. Additionally, it would be highly valuable to include relevant references on this subject. This information could be seamlessly integrated, perhaps before the last pharagraph. Also the introduction should emphasize the novelty of the research. What is really new respect the state of the art and technique and why is relevant.

- Line 33 - The abstract and the introduction focusses the interest of this research in the potential application of high moisture extrusión of meat analogue production. Some questions should be answered, and more information shouw be included about the need and the justification of this afirmation. This is tu purpose fo the introduction Section, also including some additional references:

- Why is important for the society the meat analogue production – line 31 - ?

- Why is important for the industry the high moisture extrusion of meat analogues – line 33 - ?

- Which production processes are improved by this extrusion process and why is it better – line 33- ?

- line 90 - Materials and methods. The description of materials is detailed and good but it should be clarified why water content and moisture are used as a same, or if they refer to equivalent magnitude.

- line 127 – Why is important the reological analysis of 7:3 SP-WG blend? Sometthing is wrong in this subsection as it only mentions 7:3 SP-WG blends while all blends are included in results secrtons.

- line 151 -  When describing the cooling die at the end of the extruder, shouldn´t be better to add a description of the extruded products also. Are they calibrated to a 70x15mm rectangular section straigth bar?. Photos of the extruded materiales shown in results section differ from the geoemtry described in the cooling die. There should be a clear description of the diferent specimens obtained form the HME process.

- line 156 – How is the mold to cut dog-bone specimens of 4-6 mm thickness from a 15mm thickness extruded bar in parallel and perpendicular direction?. How is the shape of the material before cutting the sepcimens.

- line 212 - In Figure 1, what is the meaning of the letters A, B, C, D and E?. Also, the correct thing to do would be to add the deviation to this bar chart.

- line 230 – In Figure 2, the indications (A), (B) and (C) should be the font type and size.

- line 327 – As it happens in Fig. 1, In Figure 7, what is the meaning of the letters A, B, C, D, AB and E?. Also, the correct thing to do would be to add the deviation to this bar chart. Why is this chart B&W chart and the previous one cloured?

- line 335 – Results about fracture stress and strain show small diferences specially in strain where all combinations evidence more or less 100% of strain at fracture. An additional discussion about these results seems to be important because , these similarities could mean that there is not a large influence betweet using one kind of protein or other, also it could be deduced that mechanical response is more influence for the presence of voids. Any additional explanation would be appreciated.

- line 347 – Permeability of extrudates. This subsection is very intersting but the text is not easy to follow. During the first pharagrapgh the text describes clearly the images but refers to figure 5, firstly, and to figures 2 and 3 finally. Is this right? What are the relationships between these figures and this subsection? . If yes, it should be described with more details.

- line 355 – Figure 9 (B). How is permeabilty quantified? In methods description only cualitative description about permeability is mentioned, but in this figure, a measure (%) of permeability is presented. In methods subsection it should be clearly defined.

-  About the conclusiones some comments are made.

* In line 237, and figures 3 and 4, it is clearly seen that the blends with SCI addition don´t follow the behaviour of blends with WP o EWP addition as expected. They are nearly the control blend or the HSPI and CP blends. It is something relevant and should be discussed in section 3 and probably, depending on the conclusions, mentioned in section 4. On the contrary, the conclusions describe the results of this blend as if they were the same as those of WP and EWP.

* In line 364, “The variations in structure and porosity directly affected the tensile properties of the extrudates, with CP demonstrating the highest anisotropic characteristics”. Checking the results, this afirmation is right and also, the results of SCI blends are closer to those obtained for WP and EWP blends. Results clearly show the increment of anisotropy and mechanical properties when adding proteins with crosslinking effect. In the results and discussion or in the conclusiones, the cause-effect of the improvements in mechanical properties should be described in Deep. Maybe a SEM (Scan Electron Microscopy) of the fracture surfaces of the blends would be of interest to improve the discussion.

* In lines 367, “These findings provide insights into the role of soluble proteins in modifying the structural and mechanical properties of SPI-WG extrudates, which have implications for their use in meat analogue production.”. The reviewer fully agree this conclusion. However, there has not been a correct definition of the requirements of a meat analogue. That is to say, it has not been clearly defined in the introduction what anisotropy or what permeability or what type of texture is intended to be achieved. It is not necessary to define it with a specific value, but it is important to define to what extent the results allow us to approximate a meat analogue. Moreover, at least in the introduction, it should have been indicated what correlation exists between the control product (7:3 SP-WG) and a meat analogue and what properties are to be achieved with the proposed blends. And therefore, in the conclusions section, it should be reflected what rate of improvement has been obtained in those quantitative variables, or to what extent there is a qualitative improvement. Not only mention that the results are related to the search for a meat analogue.

With these comments, the reviewer only intends to provide a constructive vision that would allow the paper to be substantially improved. I believe that the authors should take these comments into account in order to consider that the paper should be published. The findings are really relevant, but more emphasis needs to be placed on that extent.

Author Response

The synthesis of polymers derived from a blend of wheat and soy proteins has gathered attention in recent research. Improving these polymers by incorporating small quantities of soluble proteins is a path that has already been explored. A global investigation, as outlined in this article, is both innovative and highly relevant to the scientific community, especially in relation to the enhancement of mechanical properties and anisotropy.

The introduction is clear and the objectives well described. The last paragraph describes clearly the extruded blends and the control simple. It could be simplified to the obtention different blends made of SPI and WG with 10% of chosen proteins which share similarities with the WG hydrolysate used in a previous study. This 10% correspond to the WG fraction.

Some suggestions from the reviewer are described below:

Q1- Line 71 - The introduction should encompass a description of the requirements or properties that need to be investigated to ascertain the production of a meat analogue. Additionally, it would be highly valuable to include relevant references on this subject. This information could be seamlessly integrated, perhaps before the last paragraph. Also, the introduction should emphasize the novelty of the research. What is really new respect the state of the art and technique and why is relevant.

R1: Thanks for your suggestion. We rewritten the introduction and cited relevant references in revised manuscript (Line32-44, line77-83).

Q2- Line 33 - The abstract and the introduction focusses the interest of this research in the potential application of high moisture extrusion of meat analogue production. Some questions should be answered, and more information show be included about the need and the justification of this affirmation. This is to purpose for the introduction Section, also including some additional references:

- line 31 - Why is important for the society the meat analogue production?

- line 33 - Why is important for the industry the high moisture extrusion of meat analogues?

R2We have rewritten the introduction in revised manuscript (Line32-44).and we have already mentioned in the first sentence (Line 29-32) that the meat analogue production reduces the problems of health, ecological, environmental pollution, and social morality.

Q3- line 33-Which production processes are improved by this extrusion process and why is it better?

R3We have added the information in line 40-44: HME allows for the creation of products with a moisture content similar to real meat, which eliminates the need for a rehydration step during the production process of meat analogue compared to low moisture extrusion.

Q4- line 90 - Materials and methods. The description of materials is detailed and good but it should be clarified why water content and moisture are used as a same, or if they refer to equivalent magnitude.

R4Thanks for your valuable suggestion. We have modified the "moisture" to "water" in revised manuscript (Line 102, 104).

Q5- line 127 - Why is important the rheological analysis of 7:3 SP-WG blend? Something is wrong in this subsection as it only mentions 7:3 SP-WG blends while all blends are included in results sections.

R5We have rewritten this section (Line142-143). SPI and WG were mixed at a ratio of 7:3 based on the literature (Chiang et al., 2019). The rheological test sample consisted of SPI, WG and soluble protein in a weight ratio of 7:2:1.

References:

Chiang, J. H., Loveday, S. M., Hardacre, A. K., & Parker, M. E. (2019). Effects of soy protein to wheat gluten ratio on the physicochemical properties of extruded meat analogues. Food Structure, 19, 100102. https://doi.org/10.1016/j.foostr.2018.11.002

Q6- line 151 - When describing the cooling die at the end of the extruder, shouldn´t be better to add a description of the extruded products also. Are they calibrated to a 70x15mm rectangular section straight bar? Photos of the extruded materials shown in results section differ from the geometry described in the cooling die. There should be a clear description of the different specimens obtained from the HME process.

R6Thanks for your suggestion. We have rewritten and cited relevant reference (Line 168, 161-163). The more detail about the extruder can be seen the Deng´s paper (Qian, Deng, Chen, 2023) in our lab (Figure 1). The cross-section of the extrudate was 10 × 70 mm (H × W) rectangular. The figure 5 of the extrudates shown in the results were obtained by tearing and folding the edge portion of the extrudates.

References:

Qian, Deng, Chen, J. (2023). High-moisture extrusion of soy protein : Effects of insoluble dietary fiber on anisotropic extrudates. Food Hydrocolloids, 141(March), 108688. https://doi.org/10.1016/j.foodhyd.2023.108688

Figure 1. Schematic diagram of twin-screw extruder structure and functional partition.

Q7- line 156 - How is the mold to cut dog-bone specimens of 4-6 mm thickness from a 15mm thickness extruded bar in parallel and perpendicular direction? How is the shape of the material before cutting the specimens.

R7Thanks for your suggestion. We have cited relevant reference (Line 184). 

Figure 2. Schematic illustration of the tensile test samples taken from the extrudates using a dog-bone-shaped mold (Qian, Deng, Chen, 2023).

Q8- line 212 - In Figure 1, what is the meaning of the letters A, B, C, D and E? Also, the correct thing to do would be to add the deviation to this bar chart.

R8We changed the uppercase ABCD to lowercase abcd, letters indicating variability between samples. We have plotted the error bar on the chart. Because of the better parallelism of the data in this experiment, which causes the error line to look short, we have changed the red color instead (Figure 1).

Q9- line 230 – In Figure 2, the indications (A), (B) and (C) should be the font type and size.

Q9Thanks for your valuable suggestion. We have resized figure 2C.

Q10- line 327 – As it happens in Fig. 1, In Figure 7, what is the meaning of the letters A, B, C, D, AB and E? Also, the correct thing to do would be to add the deviation to this bar chart. Why is this chart B&W chart and the previous one colored?

R10Thanks for your valuable suggestion. We redraw the figure7.

Q11- line 335 – Results about fracture stress and strain show small differences specially in strain where all combinations evidence more or less 100% of strain at fracture. An additional discussion about these results seems to be important because, these similarities could mean that there is not a large influence between using one kind of protein or other, also it could be deduced that mechanical response is more influence for the presence of voids. Any additional explanation would be appreciated.

R11Thank you for your suggestion. The voids is spherical in extrudates (Figure.6). The AI of E-CP and E-HSPI increased was due to the increase of fracture stress in the parallel direction, rather than the decrease of fracture stress in the perpendicular direction. Thus, we believe it is rigorous to conclude that the mechanical response is influenced by both anisotropic structure of protein matrix and the presence of voids. Additional explanation was added in section 3.4.

“The increased AI for E-CP and E-HSPI was due to an increase of fracture stress in the parallel direction, rather than a decrease of fracture stress in the perpendicular direction. Thus, we deduced that the mechanical response is influenced by both the anisotropic structure of protein matrix and the presence of air bubbles.” (Line355-358)

Q12- line 347 - Permeability of extrudates. This subsection is very interesting but the text is not easy to follow. During the first paragraph the text describes clearly the images but refers to figure 5, firstly, and to figures 2 and 3 finally. Is this right? What are the relationships between these figures and this subsection? If yes, it should be described with more details.

R12Thanks for your valuable suggestion. We have rewritten this section (line370, 373).

Q13- line 355 - Figure 9 (B). How is permeability quantified? In methods description only qualitative description about permeability is mentioned, but in this figure, a measure (%) of permeability is presented. In methods subsection it should be clearly defined.

R13Thanks for your valuable suggestion. We defined the permeability in revised manuscript (Line 213-215).

About the conclusions some comments are made.

Q14* In line 237, and figures 3 and 4, it is clearly seen that the blends with SC addition don´t follow the behavior of blends with WP o EWP addition as expected. They are nearly the control blend or the HSPI and CP blends. It is something relevant and should be discussed in section 3 and probably, depending on the conclusions, mentioned in section 4. On the contrary, the conclusions describe the results of this blend as if they were the same as those of WP and EWP.

R14Thank you for your suggestion. The behavior of SC was quite special among all soluble proteins. It aggregates when temperature above 120°C, which would result in an increase of complex modulus. However, the complex modulus of blends starts to decrease at this temperature range, which dominate the complex modulus result. That is why the blends with SC addition do not follow the behavior of blends with WPI and EWP addition. Moreover, the blends with SC addition also do not follow the behavior of the control blend or the HSPI and CP blends based on the result of tan δ.

We added extra explanation in Section 3.1.3.

“Although SC showed different behavior of blends with WPI and EWP (Fig.3 and Fig. 4), they could exhibit similar impact during extrusion.” (Line277-278)

Q15* In line 364, “The variations in structure and porosity directly affected the tensile properties of the extrudates, with CP demonstrating the highest anisotropic characteristics”. Checking the results, this affirmation is right and also, the results of SCI blends are closer to those obtained for WP and EWP blends. Results clearly show the increment of anisotropy and mechanical properties when adding proteins with crosslinking effect. In the results and discussion or in the conclusions, the cause-effect of the improvements in mechanical properties should be described in Deep. Maybe a SEM (Scan Electron Microscopy) of the fracture surfaces of the blends would be of interest to improve the discussion.

R15Thank you for your suggestion. Please refer to our response in R11. It might be more effective to investigate the fracture behavior and present images of the fracture surface of the extrudates. However, selecting an appropriate tool for this purpose poses a challenge. SEM or CLSM, while providing detailed views, might be too zoomed in to offer a comprehensive representation.

Q16* In lines 367, “These findings provide insights into the role of soluble proteins in modifying the structural and mechanical properties of SPI-WG extrudates, which have implications for their use in meat analogue production.”. The reviewer fully agreed this conclusion. However, there has not been a correct definition of the requirements of a meat analogue. That is to say, it has not been clearly defined in the introduction what anisotropy or what permeability or what type of texture is intended to be achieved. It is not necessary to define it with a specific value, but it is important to define to what extent the results allow us to approximate a meat analogue. Moreover, at least in the introduction, it should have been indicated what correlation exists between the control product (7:3 SP-WG) and a meat analogue and what properties are to be achieved with the proposed blends. And therefore, in the conclusions section, it should be reflected what rate of improvement has been obtained in those quantitative variables, or to what extent there is a qualitative improvement. Not only mention that the results are related to the search for a meat analogue.

R16Thank you for your suggestion. We have modified the introduction and conclusion section.

With these comments, the reviewer only intends to provide a constructive vision that would allow the paper to be substantially improved. I believe that the authors should take these comments into account in order to consider that the paper should be published. The findings are really relevant, but more emphasis needs to be placed on that extent.

Round 2

Reviewer 2 Report

Comments and Suggestions for Authors

The reviewer expresses gratitude to the authors for their efforts in reading, analyzing, and providing responses to the questions posed.

Some suggestions from the reviewer are described below:

R1: Thanks for your suggestion. We rewritten the introduction and cited relevant references in revised manuscript (Line32-44, line77-83).

Q1: The reviewer appreciates the response and reiterates the importance of giving prominence to both the societal and scientific interest generated by meat analogues. Additionally, the reviewer finds well-justified the significance attributed to the HME process for this purpose. Also, the description of the fibrous texture creation and the relationship between the focus of the research and the penetration of marinades is well justified.

-       Line 37. It is worth noting a minor error in the inclusion of the dot "." before the citations "[4][5]"; it should be placed after the citations instead.

Q2 The reviewer understands the response to Q8 (rev1) and appreciate the addition of error bars to the chart.

- line 212 – Figure 1Shouldn´t be indicated the meaning of these lowercase abcde, letters in the text?.

- line 327 – Figure 7.  The same.

No more questions or suggestion are added

Author Response

Q1: Line 37. It is worth noting a minor error in the inclusion of the dot "." before the citations "[4][5]"; it should be placed after the citations instead.

R1Thanks for your suggestion. This error has been corrected.

Q2: - line 212 – Figure 1Shouldn´t be indicated the meaning of these lowercase abcde, letters in the text?.

 - line 327 – Figure 7.  The same.

R2Thanks for your suggestion.

“Different letters (a, b, c, d, e) express the significant differences (p < 0.05).” was added in the Fig. 1 and Fig.7.

“Different letters (a, b, c) express the significant differences (p < 0.05).” was added in the Fig. 9.
